# Impacts and Drivers of Smooth Brome (*Bromus inermis* Leyss.) Invasion in Native Ecosystems

**DOI:** 10.3390/plants11101340

**Published:** 2022-05-18

**Authors:** Rakhi Palit, Edward S. DeKeyser

**Affiliations:** School of Natural Resource Sciences, North Dakota State University, Fargo, ND 58102, USA; edward.dekeyser@ndsu.edu

**Keywords:** *Bromus inermis*, ecosystem, homogenized, invasive species, management, native, smooth brome

## Abstract

Smooth brome (*Bromus inermis* Leyss.) is an invasive cool-season grass that has spread throughout the Great Plains of North America. The species is considered one of the most widespread exotic grasses that has successfully invaded both cool-season and warm-season native prairies. In the prairies where it has invaded, there has often been a total elimination of native species and an overall homogenization of ecosystems. Smooth brome has greater competitive abilities compared to many native grasses and can foster their total elimination in many instances. The greater competitiveness can be partially attributed to its ability to alter the soil and hydrological properties of a site. It is a deep-rooted rhizomatous grass species that thrives in nitrogen-enriched soil, and since its leaf tissue decomposes faster than native species, it in turn increases the soil nitrogen level, causing positive plant-soil feedback. Moreover, smooth brome is able to transport the required nutrients from older plants to the newer progenies invading new nutrient-depleted areas, making it a potent invader. However, the impact of smooth brome is not limited to soil biochemistry alone; it also affects other ecosystem components such as the movement and behavior of many native arthropods, thereby altering the overall population dynamics of such species. Thus, smooth brome invasion poses a serious threat to the remnant prairies of the Great Plains, and efficient management strategies are urgently needed to control its invasion. Control measures such as mowing, grazing, burning, and herbicide application have been effectively used to manage this species. However, due to the widespread distribution of smooth brome across North America and its adaptability to a wide range of environmental conditions, it is challenging to translate the management strategies from one area to another.

## 1. Introduction

Agricultural land conversion has resulted in significant habitat fragmentation throughout the grasslands in the Great Plains [1,2]. Rigorous agricultural production and alterations in original disturbance regimes of grazing and fire that facilitated the formation and maintenance of native prairies have also contributed to habitat fragmentation [3]. Indeed, this land fragmentation has caused severe loss in about 90% of tall-grass, 70% of mixed-grass, and 50% short-grass prairies and has transformed prairies of the Great Plains into some of the most highly threatened and least protected ecosystems in North America [1,2,4,5,6]. Overgrazing and the suppression of fire have been deteriorating the fragments even further [7]. Thus, native species in the fragmented North American prairies are facing a multitude of challenges, including climate change, impaired dispersal, genetic bottlenecking, and competition with exotic species. Increasing anthropogenic disturbances, especially habitat fragmentation, are also making these grasslands more susceptible to invasions by exotic species [1,8].

Smooth brome (*Bromus inermis* Leyss.), belonging to the family Poaceae, is one of the most aggressive exotic grasses spreading across the Great Plains [1,3,9,10]. Smooth brome and another dominant exotic grass, Kentucky bluegrass (*Poa pratensis* L.), can make up about 62% of the exotic species cover in some Great Plains grasslands [1,11]. As in the case of other problematic plant species, smooth brome was deliberately introduced beyond its native range [12]. It was first introduced to North America in around 1880 from Hungary and Russia as a forage species for soil stabilization and retention purposes [13,14,15]. According to the historical records, this grass species was first imported in the late 1880s to the California Agricultural Experiment Station, USA, as a potential drought- and cold-tolerant forage [12,16]. It was widely cultivated and used for pastures, hay, and soil conservation [10]. The grass has also been used as wildlife cover through-out the Great Plains [10,16,17]. For example, smooth brome seeds along with seed mixtures of sweet clover (*Melilotus officinalis* (L.) Lam) and alfalfa (*Medicago sativa* L.) from the family Fabaceae have been widely used by U.S. Department of Agriculture (USDA) programs and natural resource managers for nesting cover and habitats for waterfowl, gray partridge, and ring-necked pheasants [10,18,19]. However, it escaped from cultivation and started growing alongside disturbed roads and eventually spread to native prairies across the Great Plains, where it poses threats to the surrounding native rangeland ecosystem [15,20,21,22]. Presently, smooth brome is considered one of the most widespread exotic grasses which has successfully invaded both cool- and warm-season native prairies in North America [3,10,23]. This escape and invasion of native prairies has occurred over a relatively short period of time [13,24]. Although smooth brome has been often recognized as an invasive rangeland species [22,25,26], it has not received the attention needed to restore the native prairie area remaining [13,27]. In this review, we discuss the potential factors that aid in the successful invasion and spread of smooth brome and its impacts on the native prairies.

## 2. Eco-Physiological and Environmental Attributes Driving the Invasiveness of Smooth Brome

Smooth brome is a perennial, fast-growing, C3, cool-season grass species with invasive tendencies [1,10,12]. In general, smooth brome can be easily distinguished from other *Bromus* species mainly by the presence of its non-pilose lemma and perennial rhizomes. A few other distinctive features include hairless upper leaf surfaces, lemma with purple tinge near the margin, a prominent nerve on the first glume, etc. [22]. Smooth brome usually grows in temperate regions with annual precipitation higher than 330 mm [10,28]. The ability to withstand dry environments makes this species a valuable forage and pasture crop in arid temperate regions [22]. For example, under greenhouse conditions, smooth brome demonstrated higher tolerance to moisture stress than the native grasses from the family Poaceae, *Elymus lanceolatus (*Scribn. & Smith) ssp. *lanceolatus,* and *Nassella viridula* (Trin.). Moreover, this invasive rangeland species can endure harsh winter weather, including several days of freezing temperatures [22,29].

It is a rhizomatous grass species with strong, deep roots that can grow up to 1.4 m in well-drained silt and clay-loam soil and can grow to a height of 0.76 m [1,10,16,20,22]. The prolific growth of the rhizomes is a particularly important factor determining the invasiveness of smooth brome [30] (Figure 1). Additionally, sustainable vegetative growth enhances the density and vigor of the vegetative tillers; these prolific above- and below-ground growths make this species more compatible for above-ground and below-ground competition [22,31,32]. It reproduces from mid-summer until autumn by seed production and produces panicles during late spring or early summer. Smooth brome also reproduces vegetatively via rhizomes [10,23,28]. Smooth brome commences proliferous spring growth earlier than surrounding native grasses [1] (Figure 1). As the biomass production starts earlier than the neighboring C4 grasses, this gives this species an edge to accumulate available resources with minimum competition, limiting resources for the warm-season grasses [12,33]. This invasive grass can also alter the soil properties, which provides this species higher competitive abilities over the native neighbors [15]. Thus, smooth brome forms monocultural stands over time and eventually outcompetes the native species from their original habitats [21,22]. Smooth brome is not always a favorite forage for native ungulates and domestic cattle, which prefer native-dominated areas over smooth-brome-infested areas [22,34]. Thus, the lower palatability of this grass has resulted in its increased abundance in native prairies (Figure 1). Eventually, the overgrazing of surrounding uninvaded areas produces disturbances that allow for smooth brome cover to establish [22,34,35,36,37]. However, other research has shown grazing negatively impacts smooth brome and favors natives in the Northern Great Plains, while non-use (e.g., no grazing, no fire, and no mowing) allows the grass to proliferate [20].

Smooth brome encroachment is generally greater in agricultural-land-dominated regions, where the soil is highly disturbed by farming operations [6,38]. In the fragmented prairies, deep-rooted rhizomes assist smooth brome in spreading vegetatively along the edges of the patches, whereas wind-dispersed seeds travel rapidly to open sites to establish new progenies in the disturbed soil [22,39,40]. Smooth brome employs a “frontal invasion” strategy, by which this invasive species gradually spreads from the introduction sites to more distant places [38,40,41]. This grass species has been introduced for soil reclamation and stabilization following new road construction. These roads then function as invasion corridors [22,40], and higher frequencies of smooth brome along roads in comparison to the edges of croplands could be explained by the “frontal invasion” strategy [40,42]. Smooth brome invasion such as this often alters soil and hydrological properties and successional pathways, which in turn favors the invasion process [21,43,44,45], and such an invasion by an exotic species leads to a significant decline in biodiversity [3,46].

Smooth brome thrives in nitrogen-enriched soil [1,21]. Moreover, it breaks down its leaf tissue faster than native neighbors and thus significantly increases the soil nitrogen levels [21]. This species has an excellent ability to transport required nutrients from older plants to the newer progenies invading a new nutrient-depleted range [1,39]. Thus, the productivity, dominance, and invasiveness of smooth brome increase with the augmented available nitrogen [22,32,47]. Previous studies also indicated a direct correlation between nitrogen supply and higher yield and seed production in smooth brome [21,48,49,50]. Harrison and Crawford (1941) [48] reported that nitrogen supplementation resulted in increased and larger vegetative tiller production in smooth brome [15]. The increased cover of smooth brome also increased the rate of photosynthesis and carbohydrate production by individual plants [15,21], and these plants efficiently stockpiled the surplus carbohydrate for the next growing season [15,48]. Disturbances causing small nutrient-rich (i.e., higher nitrogen) patches could contribute significantly to the growth and vigor of this species. For example, in Minnesota, smooth brome cover growing on *Formica obscuripes* (Forel)—from the family Formicidae (thatching ant)—mounds demonstrated higher growth and produced more pollen than those grown on undisturbed grasslands [22,51]. However, there is some evidence that too much nitrogen might hamper vegetative reproduction. The addition of nitrogen fertilizer decreases the root and rhizome growth, while above-ground biomass is improved [22,32,52].

## 3. Impacts of Smooth Brome on the Native Ecosystem

Invasive species significantly influence ecosystem functioning and integrity, which poses major threats to the surrounding native plant communities [53,54,55]. Actually, invasive species, such as smooth brome, are considered to be key factors, along with habitat fragmentation, that contribute to the extinction risk of native plants [56,57,58]. Smooth brome exhibits a significantly higher competitive ability compared to native grasses and promotes their eradication in an invaded area. For example, Dillemuth et al. (2009) [56] found detrimental effects of smooth brome on the growth, survival, and extinction rate of the native grass species, prairie cordgrass (*Spartina pectinata* (Bosc ex Link), family Poaceae). In smooth-brome-dominated patches, cordgrass colonization was about 1.3 times less, and the probability of the eradication of cordgrass was 8 times higher. In a greenhouse study, smooth brome proved its competitive ability against five native grass species under different moisture regimes [56,59]. Moreover, it has been shown that the native forb species pasque flower (*Pulsatilla patens* (L.) Mill.) from the family Ranunculaceae has decreased in growth where smooth brome has invaded [13,26]. Because smooth brome can invade native grasslands and form dense monocultural stands by supplanting and replacing native species (Figure 2), it could eventually transform a diverse ecosystem dominated by native species into a homogenized, novel ecosystem [4]. A significant part of the remnant prairies in the Great Plains are affected by smooth brome invasion, and efficient management strategies are needed to control this species [3,60].

The apparent changes in the above-ground plant community structure and dynamics caused by invasive species suggest significant below-ground changes, such as the distribution of roots in the soil, impacts on soil microbes, and changes to other soil properties and elements [61,62,63,64,65], which eventually influence ecosystem processes, including biogeochemical cycles [44,65,66,67,68]. Through the changes in plant–soil feedback, the structure and diversity of the plant community significantly affect the nitrogen cycling rates [65,69,70,71,72]. The composition of the above-ground community strongly influences the quality, amount, and decomposition rates of the added litter to the soil [65,73,74]. Moreover, the quality and quantity of the litter in the invaded ranges could be substantially different in native-species-dominated areas, which would ultimately amend the mineralization rates [44]. Altered below-ground root composition caused by plant invasions potentially changes the quality and quantity of root-exudates, which significantly influence the composition and richness of the soil microbial community [63,65,75,76,77,78,79]. An invasive species can alter the soil nitrogen cycle either directly by itself (e.g., through nitrogen use efficiency, residence time, etc.) or indirectly by interacting through soil microbes (e.g., influencing the abundance of nitrifying microbial communities) [65,80]. A previous study reported the overall effects of exotic species on soil nitrogen cycling, elevations in total nitrogen content, mineralization rates, and changes in AOA (ammonia-oxidizing archaea) and AOB (ammonia-oxidizing bacteria) on smooth-brome-dominated areas compared to native fescue prairie [65]. Higher above-ground and below-ground biomass produced by smooth brome might have caused the gross increased mineralization rates in that study. Moreover, as smooth brome encroachment did not dislocate any particular native species at the study site, this suggests that these changes in the nitrogen cycle were potentially triggered by increased smooth brome abundance rather than declined native species richness [65,81]. Future studies should explore if smooth brome invasion also causes changes in denitrifying communities and denitrification processes, which are responsible for the emission of nitrous oxide, a potent greenhouse gas. Such findings will help us understand the dynamics of smooth brome invasion in the era of global climate change.

Smooth brome invasion directly and/or indirectly impacts different trophic levels of the native ecosystem as well. For instance, primary consumers of cordgrass, a herbivore planthopper *Prokelisia crocea* (Van Duzee), from the family Delphacidae, and its parasitoid *Anagrus columbi* (Perkins), from the family Mymaridae, both have considerably higher chances of migration from cordgrass patches surrounded by smooth brome than those patches embedded in a native species matrix [25,56]. Consequently, the densities of both *Prokelisia crocea* and *Anagrus columbi* declined by up to 50%. Furthermore, their extinction risks are 4–5 times higher in brome-dominated areas over the cordgrass patches adjacent to the native species [25]. Thus, smooth brome invasion could lead to the extinction of both the planthopper and its parasitoid at the landscape level [82].

## 4. Potential Management

In the Northern Great Plains, smooth brome frequency was found to be higher on Federal Lands that were not managed by livestock grazing or periodic fire events, in contrast to grazed private pastures [20]. In general, in the Northern Great Plains, other lands that are not managed with grazing or burning show a similar trend of smooth brome invasion [1,6,20,83]. Control measures such as mowing, grazing, and burning have been employed to control smooth brome and Kentucky bluegrass. Timely cutting to take advantage of low root carbohydrate levels can cause damage to smooth brome [84,85]. Grazing seems to be the better management approach over burning or herbicides for controlling smooth brome without damaging the neighboring native cool-season grasses [3,86].

Fire is considered as an efficient management tool to combat smooth brome. Annual prescribed fire, especially early spring and late fall burning over several years, proved to be an effective control strategy for smooth brome, although repeated burning could significantly damage the native plant community [3]. During tiller elongation in spring, carbohydrate levels are low in plants, and it is noteworthy that previous studies reported that prescribed spring burning resulted in about a 50% decrease in the tiller density in smooth brome [3,33,87]. When an invasive cool-season grass grows with warm-season native grasses, spring burning provides a competition balance shift towards the natives and favors their growth [3,88]. There was also a decline in smooth brome density after fall burning compared to the untreated control plots [3]. Following the fall burning, native species cover increased substantially, whereas smooth brome cover remained comparatively steady over the next three growing seasons. Although burning did not destroy all the established smooth brome cover, it damaged the viable seeds on the soil surface, thus restricting smooth brome’s germination [3,89].

Many chemical herbicides, including imazapyr, imazapic, and sulfometuron, have been used to manage smooth brome. Previous studies reported that the use of these chemicals caused about a 70% decline in smooth brome productivity [3,90]. In smooth-brome-dominated grasslands, atrazine application in late April efficiently reduced smooth brome cover by up to 91% and actually transferred the cool-season dominated grassland to a warm-season dominated prairie [91,92]. The use of herbicides has shown promising results in the initial decline in the exotic cover, allowing managers to integrate other potential approaches to mitigate future invasions and restore native prairie [43]. Bahm et al. (2011) [43] also indicated that a substantial increase in native grass cover was reported in herbicide-treated plots at the end of the second and third growing seasons; however, overall native grass cover remained about the same in untreated plots. Other research has shown that the application of imazapyr [90], imazapic [93,94,95], and sulfosulfuron [93] in Kentucky, Nebraska, and South Dakota [43] improved native grass cover. These native species might be highly competitive against smooth brome, resulting in the stunted tiller growth of this invasive species and controlling its overall growth and spread [92,96]. In a native prairie restoration study, applying glyphosate before inter-seeding with native species significantly decreased smooth brome biomass and substantially increased the species richness of native warm-season grasses [97]. Therefore, the application of herbicides could be efficiently used for both combating smooth brome invasion and increasing native cover.

The timing of application of the management tools plays a key role in managing native prairies [60,98,99]. For example, Brueland et al. (2003) [100] suggested that after the plants get even one fully collared leaf per vegetative tiller, grazing would not damage smooth brome plants. Additionally, after the plants arrive at the elongation and reproductive stages, grazing is not an ideal management tool [99,101]. Willson and Stubbendieck (1997) [102] concluded that the elongation stage is the ideal stage to employ management strategies including grazing and fire. Prescribed burning would give optimal results during the five-leaf stage for smooth brome, which indicates the onset of the elongation stage [92,99]. Fire during this stage exerts the highest damage to the reserved carbohydrate for their survival in some areas of the Great Plains. Similarly, high-intensity grazing for a short duration during the elongation stage is detrimental to smooth brome survival and spread [12]. In a reclaimed coal mine, horse grazing was employed during the tiller elongation, and the biomass production in smooth brome was about 1/5 times lower than the untreated control plots [12,86]. Thus, timely applications of fire and grazing could be employed as effective management strategies in smooth-brome-invaded prairies.

## 5. Conclusions

Smooth brome has been widely used as a forage and pasture crop, for erosion control, phytoremediation, the reclamation of open mines, and as a soil stabilizer in areas damaged by forest fires across North America [12,22]. Several smooth brome cultivars have been developed for improved specific qualities, better adaptability to particular geographic areas, and sustaining dominance even in undisturbed areas [12,22,65,103]. The extensive distribution of this species in North America and its adaptability to a wide range of environmental and edaphic conditions make it challenging to translate efficient management approaches from one region to another. Thus, it is likely that smooth brome populations are genetically different across the continent [12]. Comprehensive understanding of the biological and ecological strategies of smooth brome is critical in achieving the optimal management strategies for this invasive species [12]. Future research should focus more on the interactions of smooth brome with surrounding biotic and abiotic factors and the underlying mechanisms of invasion and dominance of smooth brome in native prairies.

## Figures and Tables

**Figure 1 plants-11-01340-f001:**
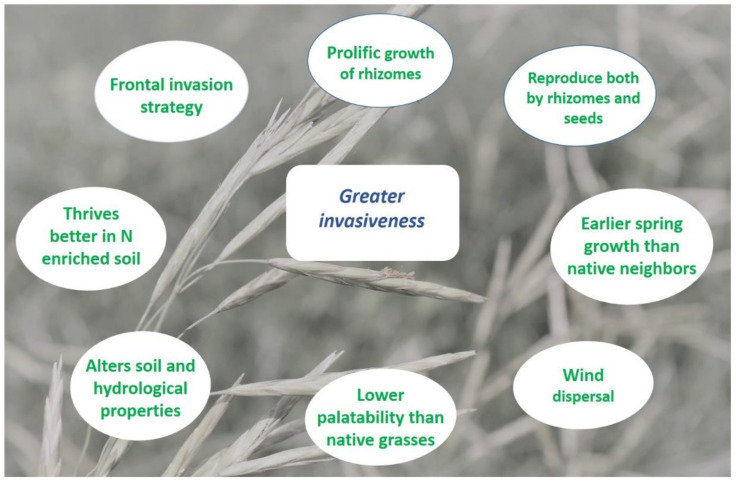
Eco-physiological factors driving the invasiveness in smooth brome. Background image was taken by Edward DeKeyser.

**Figure 2 plants-11-01340-f002:**
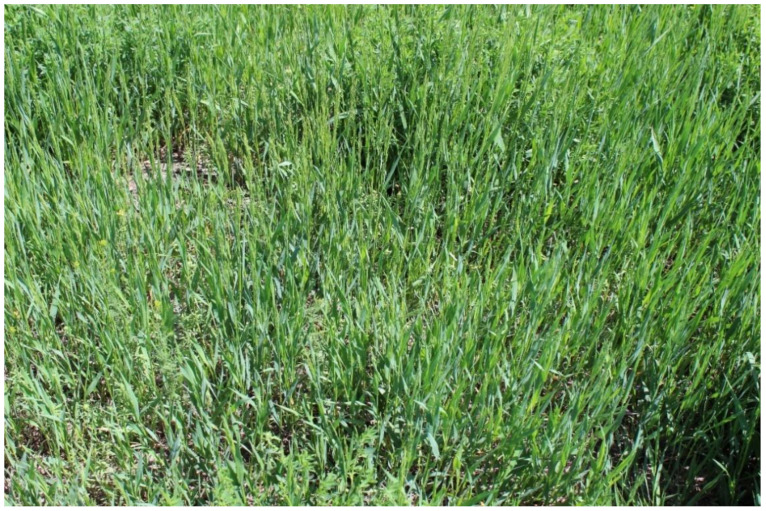
Smooth brome monocultural stands displaced native grasses in a native rangeland in Northern Great Plains. Photo was taken by Edward DeKeyser.

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
