# Peer review of "Impacts and Drivers of Smooth Brome (Bromus inermis Leyss.) Invasion in Native Ecosystems"

_plants, 2022, doi:10.3390/plants11101340_

Round 1

Reviewer 1 Report

The article gives an interesting account on the traits that perpetuate the invasion from Bromus inermis. Understanding the traits is important for management actions required to stem the growing population and its associated impacts. This manuscript further does an excellent job in providing the recommendations for biodiversity managers, in order to reduce the impacts of Bromus inermis.

Author Response

Please refer to the attached word file, Responses to the Reviewer 1.

Reviewer 2 Report

The review by Palit and DeKeyser gives an exhaustive overview about Bromus inermis and elements that contribute to its invasiveness and impacts. This is an informative contribution useful to take stock about this plant which can be a relevant threat to native species and habitats.

In my opinion, the review could be even more complete and informative if a paragraph dedicated to the description of the plant would be added, reporting taxonomy, key features to identify the plant, possible issues in identifying it. Reproduction strategies are cited in the second chapter, but maybe a brief general description (with phenology) would be necessary.

Below several specific observations:

Line 12 "eradication" is usually used to indicate a management objective (e.g. eradication programmes of alien plants), while in this case I would find "disappearance" or similar terms more adequate.

Line 44 "is" is twice repeated, and then I suggest to add the family of Bromus inermis, if you do not decide to add an initial descriptive paragraph about the species.

Line 48 please, indicate which is the native range of Bromus inermis, where it originally came from.

Line 55 add scientific name and family to sweet clover and alfa alfa

Line 75 add authors and family to Elymus lanceolatus ssp. lanceolatus and Nassella viridula. Are these species relevant because they share the same habitat of Bromus inermis? Please specify it. Then, in the review you use to indicate species with their common name, so I recommend coherently adding their common name or changing all plant common names to scientific ones.

Lines 82-83 sorry, it is not very clear to me if you are saying that this species is more compatible for above or below-ground competition and why.

Lines 83-84 please, treat separately sexual and vegetative reproduction with relative phenology.

Line 91 "outcompetes native species in their original habitats" or "substitutes" would be more adequate of "eradicates" (see my comment to line 12).

Line 92-94 Please add one or more references or indicate if it is consideration coming from personal observations.

Figure 1 Should the centrality of "greater competitive ability over native grasses" indicate that all "surrounding" traits contribute to it? Because I'm not sure that you demonstrated the greater competitive ability of Bromus inermis against all the traits (e.g. wind dispersal). So surely they are the "Eco-physiological factors driving the invasiveness in smooth brome", but I would reconsider a little bit the structure of the figure.

Line 129 add authors and family to Formica obscuripes.

Line 140 exhibits a significantly higher competitive ability compared to "several" native grasses.

Lines 143 and 147 add authors and family to Spartina pectinata and Pulsatilla patens.

Line 174 I suggest to indicate AOA and AOB also in the extended form.

Line 186 add authors and family to Prokelisia crocea and Anagrus columbi

Line 215 you cite seeds, so information about seed germinability, soil seed bank viability (long-lasting soil seed bank or not?), abundance would be very interesting and relevant to the management of the plant.

Author Response

Please refer to the word document, responses to the Reviewer 2.
